# Protein denaturation at the air-water interface and how to prevent it

**Edoardo D'Imprima[1][†]\*, Davide Floris[1][†], Mirko Joppe[2], Ricardo Sánchez[3], Martin Grininger[2], Werner Kühlbrandt[1]**

[1]Department of Structural Biology, Max Planck Institute of Biophysics, Frankfurt, Germany; [2]Buchmann Institute for Molecular Life Sciences, Institute of Organic Chemistry and Chemical Biology, Goethe University Frankfurt, Frankfurt, Germany; [3]Sofja Kovalevskaja Group, Max Planck Institute of Biophysics, Frankfurt, Germany

**Abstract** Electron cryo-microscopy analyzes the structure of proteins and protein complexes in vitrified solution. Proteins tend to adsorb to the air-water interface in unsupported films of aqueous solution, which can result in partial or complete denaturation. We investigated the structure of yeast fatty acid synthase at the air-water interface by electron cryo-tomography and single-particle image processing. Around 90% of complexes adsorbed to the air-water interface are partly denatured. We show that the unfolded regions face the air-water interface. Denaturation by contact with air may happen at any stage of specimen preparation. Denaturation at the air-water interface is completely avoided when the complex is plunge-frozen on a substrate of hydrophilized graphene.

DOI: https://doi.org/10.7554/eLife.42747.001

\*For correspondence:
edoardo.dimprima@biophys.mpg.
de

[†]These authors contributed equally to this work

## Introduction

In the short time since the resolution revolution (*Kuhlbrandt, 2014*), single-particle electron cryo-microscopy (cryo-EM) has developed into a main technique for high resolution structure determination of proteins (*Bai et al., 2015a*). To achieve high contrast and high resolution in cryo-EM, a small volume of protein solution is applied to an EM support grid (*Cheng et al., 2015*; *Passmore and Russo, 2016*) and blotted before vitrification by plunge-freezing in liquid ethane (*Dubochet et al., 1988*; *McDowall et al., 1983*). During this process, the protein is inevitably exposed to the atmosphere at a high surface-to-volume ratio. It has often been suggested that the air-water interface is a hostile environment for proteins (*Glaeser and Han, 2017*; *Ramsden, 1994*; *Taylor and Glaeser, 2008*; *Trurnit, 1960*; *Yoshimura et al., 1994*). Numerous studies from the first half of the 20th century (reviewed by Neurath and Bull (*Neurath and Bull, 1938*)) have shown that globular proteins applied to dilute buffers will eventually form an insoluble monolayer of ~10 Å thickness at the air-water interface, which means that they denature completely. A recent systematic investigation by electron cryo-tomography (cryo-ET) of 31 different proteins on cryo-EM grids has shown that all have a more or less pronounced tendency to adhere to the air-water interface (*Noble et al., 2018*). Even if the proteins do not denature, adsorption often results in preferential orientation, which is undesirable for image processing and high-resolution structure determination. The standard method of preparing cryo-EM specimens by plunge-freezing of thin, unsupported layers of protein solution is therefore potentially problematic. Recent studies (*Glaeser, 2018*; *Glaeser and Han, 2017*; *Han et al., 2017*) have drawn attention to the effects of the air water interface on proteins in solution, in particular on their integrity and orientation on cryoEM grids.

Proteins diffuse from the bulk phase of a thin layer of aqueous solution to the air-water interface in a millisecond or less (*Israelachvili, 2011*; *Naydenova and Russo, 2017*; *Taylor and Glaeser, 2008*), so that each protein can make thousands of contacts with the atmosphere during the few

seconds it takes to prepare a cryo-EM grid. At each encounter, the protein is at risk of partial unfolding. Fast nanodispensers (*Jain et al., 2012*) in combination with self-blotting grids (*Wei et al., 2018*) have been developed to minimize protein exposure to the air-water interface, and initial results look promising (*Dandey et al., 2018*; *Scapin et al., 2018*; *Xu et al., 2018*). Attempts to overcome preferential orientation include saturation of the surface with surfactants, such as fluorinated detergents (*Popot, 2010*) that interact poorly with the protein (*Blees et al., 2017*; *Efremov et al., 2015*). These approaches require careful screening or access to a sophisticated (and costly) apparatus that is not universally available. As a simpler and potentially more general solution, we propose to use a physical support that largely prevents protein contact with, and consequently denaturation at, the air-water interface.

Continuous thin layers of amorphous carbon help to spread proteins evenly on cryo-EM grids (*Bai et al., 2013*; *D'Imprima et al., 2017*; *Nguyen et al., 2015*; *Schraidt and Marlovits, 2011*). Amorphous carbon is, however, far from ideal as a support film for cyro-EM because it adds background and conducts electrons poorly (*Brink et al., 1998*; *Larson et al., 2011*). Beam-induced movement is more severe on amorphous carbon support films than on unsupported films of vitrified solutions (*Russo and Passmore, 2014*).

In contrast to amorphous carbon, graphene, a monomolecular layer of crystalline carbon, has a number of desirable properties. It is the thinnest and strongest material known and at the same time an excellent conductor (*Geim and Novoselov, 2007*). It is stable under a 300 kV electron beam (*Sader et al., 2013*), and almost completely electron-transparent to 2.13 A° resolution (the position of the first Bragg peak) and beyond (*Pantelic et al., 2012*). The main problem of graphene for cryo-EM is its extreme hydrophobicity. For an even spread of the protein solution, the graphene surface has to be made hydrophilic. Graphene oxide is less hydrophobic than graphene, but more difficult to apply to EM grids as a monolayer (*Boland et al., 2017*). Graphene can be rendered hydrophilic by plasma etching (*Russo and Passmore, 2014*) or non-covalent chemical doping, exploiting the π-π stacking interaction between graphene and aromatic planar compounds such as 1-pyrenecarboxylic acid (*Pantelic et al., 2014*). For cryo-EM, advantages of non-covalent doping include (i) the graphene structure is preserved; (ii) surface charge can be selectively modified; (iii) the number of adsorbed particles per unit area can be tuned by adjusting the concentration of the doping chemical.

We explored the denaturing effect of the air-water interface on fatty acid synthase (FAS) from *Saccharomyces cerevisiae* as an example of a large protein complex, and devised a way to avoid it. The structure of FAS is well-characterized by protein crystallography (*Jenni et al., 2007*; *Johansson et al., 2008*; *Lomakin et al., 2007*) and cryo-EM (*Gipson et al., 2010*). This makes it easy to detect and analyze which part of the complex contacts the interface and to what extent it is denatured. We use cryo-ET to locate the FAS particles on cryo-EM grids and visualize the denaturation of individual protein complexes in contact with the air-water interface. Finally, we demonstrate by high-resolution single-particle cryo-EM that a stable substrate of hydrophilized graphene avoids the denaturation of FAS complex during cryo-EM specimen preparation completely.

## Results

### Fatty acid synthase is intact prior to cryo-EM grid preparation

The FAS complex used for cryo-EM data collection was pure and homogeneous, as shown by size exclusion chromatography, SDS-polyacrylamide gel electrophoresis and blue-native polyacrylamide gel electrophoresis (*Figure 1—figure supplement 1A–C*). Thermal shift assays indicated that the complex was stable (*Figure 1—figure supplement 1D*), and at 1500–3000 mU/mg it was enzymatically fully active (*Fichtlscherer et al., 2000*; *Oesterhelt et al., 1969*; *Wieland et al., 1979*). Negative-stain EM of freshly purified FAS samples indicated that the complex was structurally intact (*Figure 1—figure supplement 2A*). Cryo-EM of the same samples in plunge-frozen, unsupported thin layers of vitrified solution on holey carbon film revealed that around 90% of the particles had suffered major structural damage (*Figure 1*). FAS particles in two-dimensional (2D) and in particular three-dimensional (3D) classification lacked between one third and one half of their density or had weak density at the distal part of the beta-domes (*Figure 1A,B*). A reconstruction of ~8000 particles was limited to 9.5 Å resolution, according to the gold-standard 0.143 FSC criterion (*Scheres and*

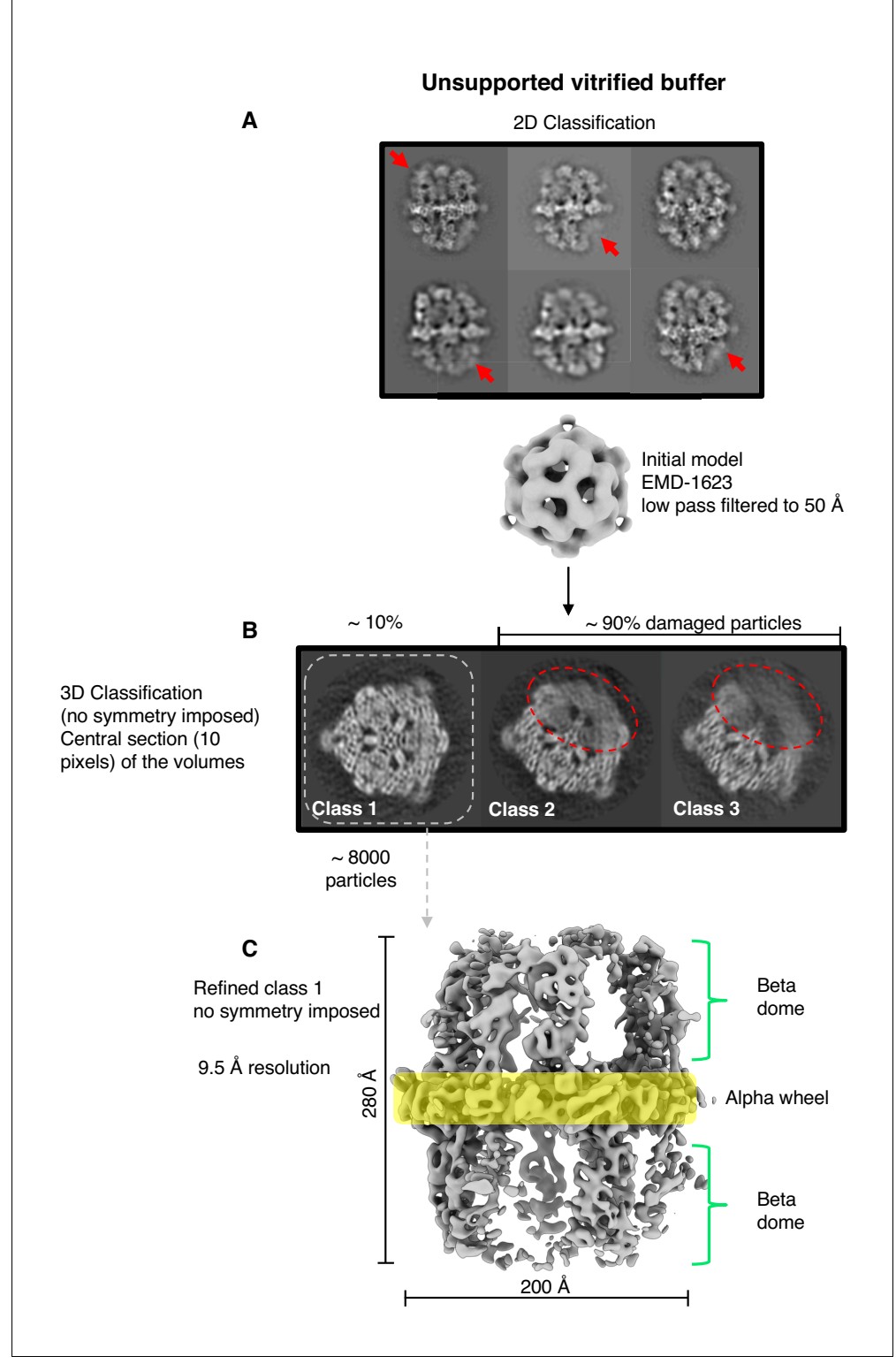

**Figure 1.** Single-particle cryo-EM results from unsupported vitrified buffer. (**A**) Two-dimensional classification of particles shows weak or absent density of beta-domes (red arrows). (**B**) The alpha-wheel structure reveals major damage to about 90% of particles (dashed red). The remaining ~10% (dashed grey) contributed to a reconstruction (**C**) at 9.5 Å resolution.

DOI: https://doi.org/10.7554/eLife.42747.002

The following figure supplements are available for figure 1:

*Figure 1 continued on next page*

*Figure 1 continued*

**Figure supplement 1.** Purification and stability of yeast FAS.
DOI: https://doi.org/10.7554/eLife.42747.003
**Figure supplement 2.** Comparison of FAS in negatively stained and unsupported cryo-EM specimens.
DOI: https://doi.org/10.7554/eLife.42747.004

*Chen, 2012*) (*Figure 1C*). Back-tracking of incomplete particles in the 3D classes (*Figure 1B*) revealed major structural defects of the protein complexes in the raw micrographs (*Figure 1—figure supplement 2B,C*). These observations led us to conclude that the protein must have been damaged prior to or during cryo-EM grid preparation.

## Particle distribution in vitrified cryo-EM grids

Next, we performed cryo-ET on the vitrified specimens prepared for single-particle cryo-EM. Several batches of purified FAS plunge-frozen by different users under different conditions were examined. All experiments indicated damaged FAS complexes in all imaged areas (*Figure 2A*). In most instances, small fragments of denatured FAS were found in the areas surrounding individual complexes (red arrows in *Figure 2A*, *Figure 2—video 1*).

Cryo-ET revealed that FAS adhered to the two opposite surfaces of the unsupported thin layers of vitrified buffer. One surface, which we refer to as the lower meniscus, was densely packed with adsorbed protein complexes. The opposite surface (the upper meniscus) had only a small number of particles attached (*Figure 2B* and *Figure 2—figure supplement 1*). Together with small ice crystals from atmospheric contamination on the outside surface of the vitrified layer, the FAS complexes on the upper and lower meniscus allowed us to trace the air-water interface exactly (*Figure 2B*).

Tomographic volumes suggested that nearly all the FAS particles in contact with the air-water interface were damaged. The particles were mostly flattened on one side and appeared incomplete (*Figure 2C*). The flattened regions aligned with the plane of the air-water interface. Particles attached to the lower and upper meniscus appeared to be equally affected, although the small number of particles on the upper meniscus precluded a statistically significant analysis. Our observations thus suggest that at some point during cryo-specimen preparation, the large majority of FAS complexes encountered the air-water interface, attached to it, and the air-exposed side unfolded before vitrification.

## Orientation of damaged FAS particles on the air-water interface

FAS particles at the air-water interface were investigated by subtomogram averaging (STA). A set of 1724 subvolumes was manually selected, and a subset of 20 randomly picked volumes was used as a reference for initial alignment. No symmetry constraints were applied. The final reconstruction indicated that one side of the FAS map lacked density, whereas the opposite side of the complex appeared complete (*Figure 3A*). FAS attached with its long axis parallel to the air-water interface, which accounts for the scarcity of top views in the single-particle analysis. To determine the orientation of the partly denatured FAS complexes relative to the air-water interface, we fitted a surface through the centers of all particles (*Figure 3B*). We then calculated the vectors pointing from the center of a complex towards its flattened side (*Figure 3C*). Finally, we assessed by how much the vectors diverged from the normal of the previously calculated plane through all particles at that position, and whether they pointed toward the air-water interface or away from it. This analysis indicated clearly that the vectors pointed towards the air-water interface (*Figure 3D*).

The structural heterogeneity of the partly denatured FAS complexes was examined by multi-reference alignment. In line with the single-particle results (*Figure 1B*), we found different degrees of particle damage. About 86% were extensively damaged, with one third or even half of the characteristic quaternary FAS structure weak or absent (*Figure 3—figure supplement 1A*). The remaining 14% had poorly resolved densities (*Figure 3—figure supplement 1B*), suggesting that even those particles on the air-water interface that appeared intact had suffered some damage. The set of subvolumes probably contained a small number of undamaged particles from the bulk phase, but visual inspection of the tomographic volumes did not reveal any. We conclude that most if not all particles

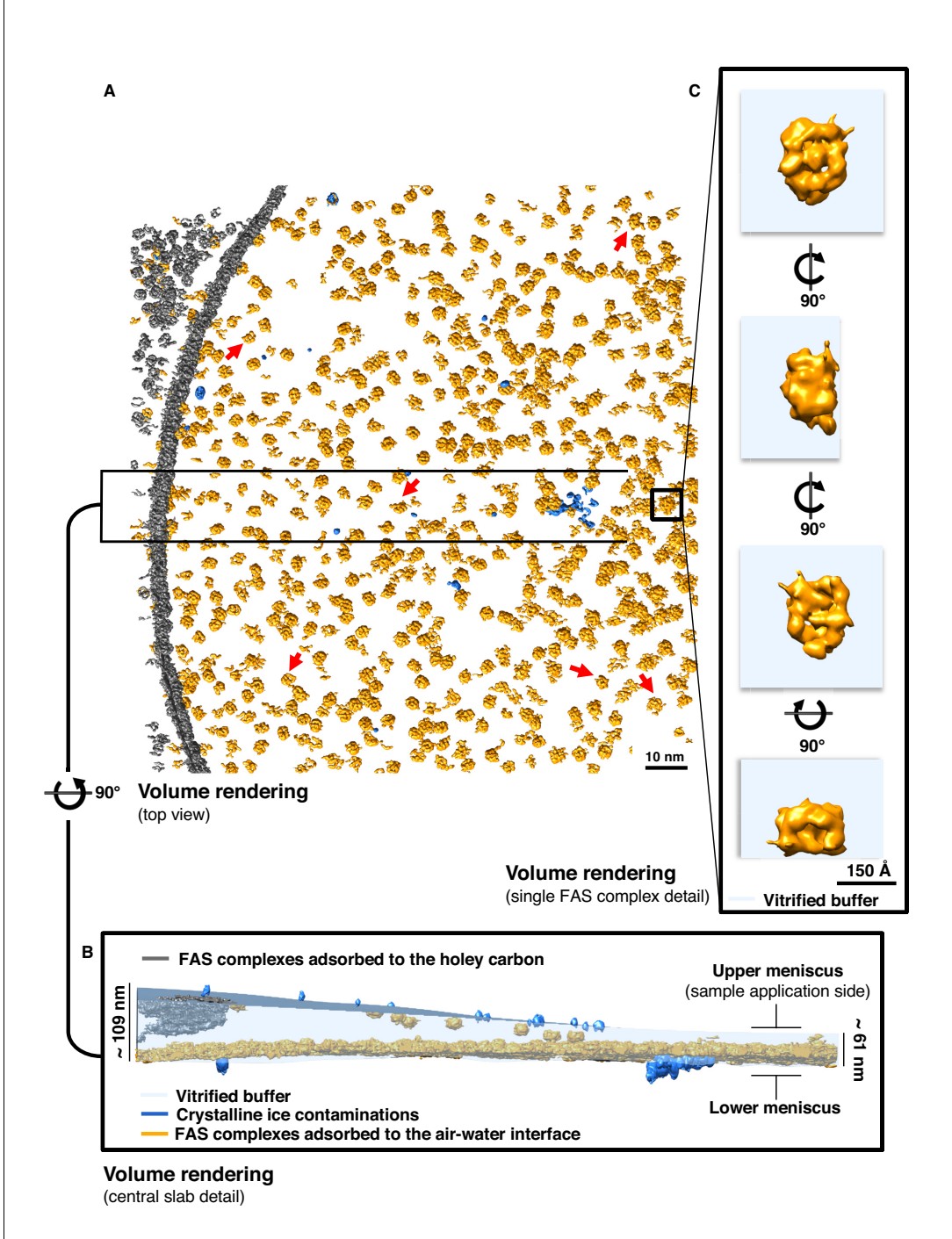

**Figure 2.** Particle distribution and structure of FAS in unsupported vitrified buffer. (**A**) Segmentation of a typical Quantifoil R2/2 grid hole with FAS complexes. Red arrows indicate fragments of FAS particles. (**B**) Slab of vitrified buffer, delimited by carbon and small contaminating ice crystals. (**C**) Detail of a single FAS complex showing morphological differences between sides facing the air-water interface or away from it.

DOI: https://doi.org/10.7554/eLife.42747.005

The following video and figure supplement are available for figure 2:

**Figure supplement 1.** Slices through a tomogram of a Quantifoil hole with unsupported vitrified solution.

DOI: https://doi.org/10.7554/eLife.42747.006

**Figure 2—video 1.** Three-dimensional rendering of FAS in unsupported vitrified solution.

DOI: https://doi.org/10.7554/eLife.42747.007

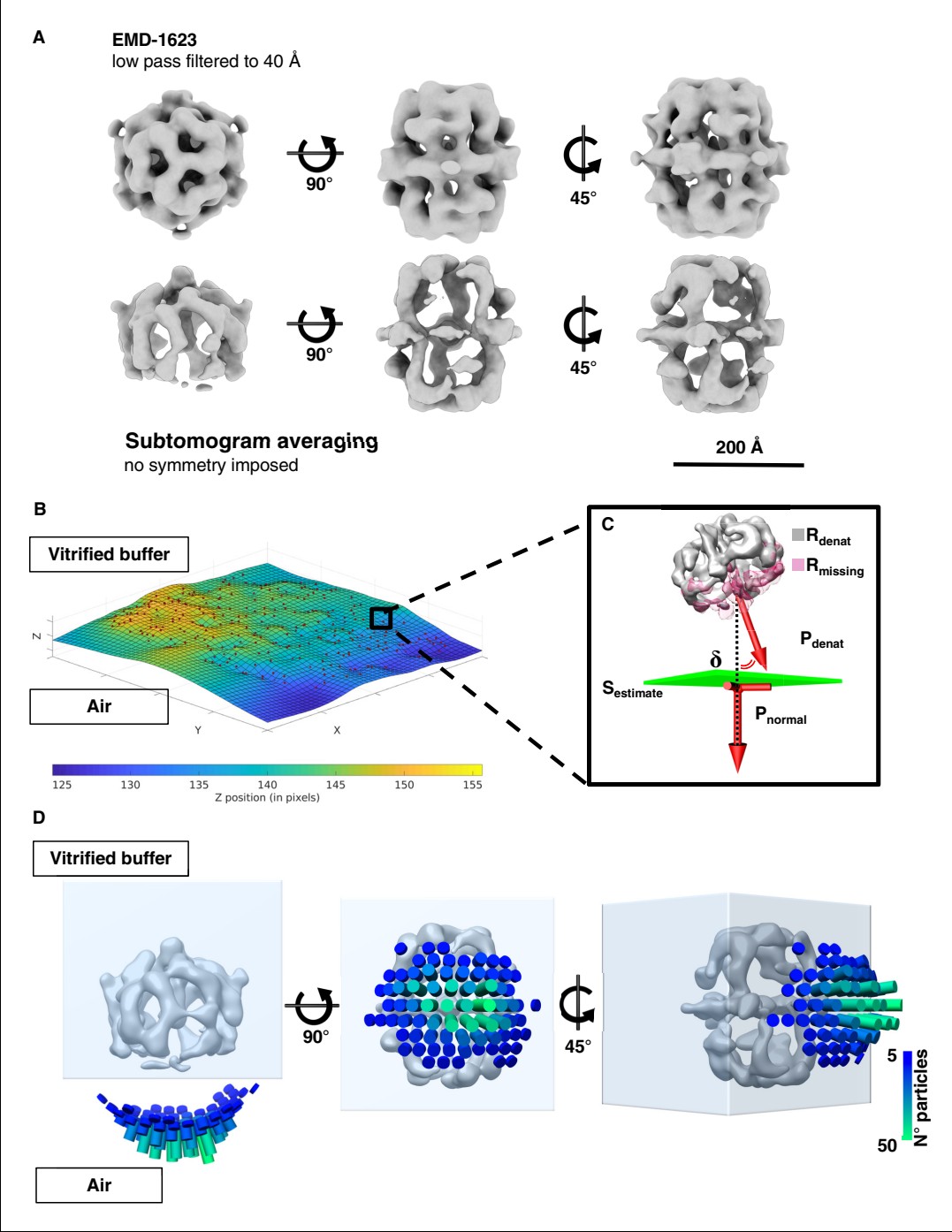

**Figure 3.** Sub-tomogram averaging and orientation of denatured FAS in unsupported vitrified buffer. (A) Subtomogram averaging confirms localized denaturation of FAS. The published cryo-EM structure of intact FAS (*Gipson et al., 2010*) (above) is shown for comparison. (B) Surface ($S_{estimate}$) through the center of all FAS complexes in the selected area. (C) Vector $P_{denat}$ describing the orientation of denatured FAS ($R_{denat}$), of the missing density ($R_{missing}$) and the perpendicular direction ($P_{normal}$) relative to $S_{estimate}$. The displacement angle is $\delta$. (D) Angular distribution of $\delta$ for all particles in reconstructed tomograms.

DOI: https://doi.org/10.7554/eLife.42747.008

The following figure supplement is available for figure 3:

**Figure supplement 1.** Multi-reference alignment of FAS in unsupported vitrified solution.

DOI: https://doi.org/10.7554/eLife.42747.009

at or near a meniscus were damaged to a greater or lesser extent by contact with the air-water interface.

## Air exposure induces protein denaturation

In a series of three experiments, we tested different ways in which exposure to air could cause protein denaturation. As before, negative-stain EM confirmed that the particles were initially undamaged (*Figure 4A*). In one experiment, we bubbled air through the sample to maximize air contact. In another experiment, we poured the protein solution over a glass rod (*Trunit, 1960*) to expose a continuous thin aqueous film to the atmosphere (*Figure 4B*). In the third experiment, we applied a 20 µl volume of FAS solution to a standard EM support grid coated with continuous carbon, and then touched the top of the droplet with a second carbon-coated grid (*Figure 4C*). In this way, we separated the particles adsorbed to the air-water interface from those adsorbed to the carbon film (*Figure 4D*). The result of each experiment was then examined by negative-stain EM (upper panels in *Figure 4B–D*). Bubbling air through the sample (experiment 1) completely denatured all FAS complexes (not shown), whereas in experiments 2 and 3 a small proportion remained intact. Denatured proteins were a predominant feature in all the three conditions except that particles adsorbed to the carbon film in experiment 3 (*Figure 4D*) were apparently undamaged. These results show that FAS at the air-water interface is denatured, whereas it remains intact when adsorbed to a solid substrate in liquid.

## Hydrophilized graphene-coated grids prevent denaturation at the air-water interface

To find out whether adsorption to a continuous support film would prevent damage also under cryo-conditions, we prepared FAS on EM-grids coated with a layer of graphene rendered hydrophilic with 1-pyrene carboxylic acid (1-pyrCA). To assess the quality of the graphene, all grids were examined by electron diffraction before vitrification. Sharp diffraction spots indicated flat monolayers of graphene (*Figure 5—figure supplement 1A,B*). The hydrophobic nature of the untreated graphene film was apparent from the repulsion of a water droplet pipetted onto the grid (*Figure 5—figure supplement 1C*). The same grids were then chemically doped with a solution of 1-pyrCA, which did

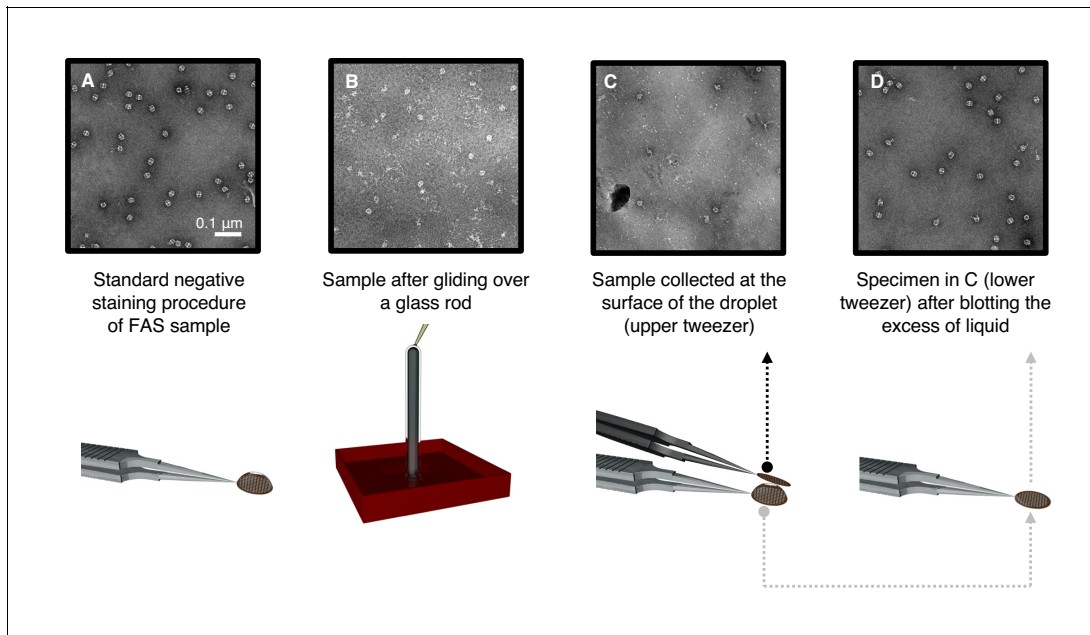

**Figure 4.** Denaturation by controlled exposure to air as analyzed by negative-stain EM. (**A**) Untreated FAS sample (control). (**B**) A thin film of FAS solution flowing over a glass rod. Most complexes are denatured. (**C**) Denatured FAS particles picked up from the top of the droplet. (**D**) Undamaged FAS particles adsorbed to amorphous carbon at the opposite drop surface.
DOI: https://doi.org/10.7554/eLife.42747.010

not degrade the crystalline order of the graphene layer (*Figure 5—figure supplement 1D,E*). The hydrophilic character of the 1-pyrCA-doped graphene was indicated by the reduced contact angle of a water droplet on the grid (*Figure 5—figure supplement 1F*). The FAS solution was applied as before, and grids were blotted and plunge-frozen as for unsupported vitreous films. The hydrophilized graphene/FAS grids were then used for cryo-ET and single-particle cryo-EM.

Cryo-ET indicated the position of the air-water and graphene-water interfaces by atmospheric ice crystals and small patches of contaminants (*Figure 5A,B*). When the graphene layer was rendered hydrophilic by 1-pyrCA, FAS had a strong preference for the graphene-water interface over the air-water interface (*Figure 5B*, *Figure 5—figure supplement 2* and *Figure 5—video 1*).

To investigate the state of preservation of FAS on hydrophilized graphene, we hand-picked a set of 2090 subvolumes and performed subtomogram averaging and multi-reference classification as for unsupported vitrified samples. Reconstructions both before (*Figure 5C* and *Figure 5—figure supplement 3*) and after (*Figure 5—figure supplement 4*) multi-reference alignment indicated that all particles were intact. The best sub-tomogram averages yielded maps at 24.6 Å and 17.1 Å resolution before and after masking (*Figure 5—figure supplement 5*). Since few if any particles stuck to the air-water interface and multi-reference alignment did not reveal any damage, we conclude that FAS does not denature on hydrophilized graphene.

## Hydrophilized graphene is suitable for high-resolution cryo-EM

To find out whether hydrophilized graphene films are suitable for high-resolution structure determination, we analyzed FAS on these grids by single-particle cryo-EM. Typical micrographs recorded at 0.9 μm defocus showed well-preserved particles (Fig. *Figure 6—figure supplement 1A*), although the background is not completely transparent. Presumably, the pristine graphene surface became contaminated to some extent with atmospheric hydrocarbons during specimen preparation, and the high dopant concentration may contribute to some loss of contrast. These factors may compromise the detection and alignment of particles that are significantly smaller than yeast FAS. The 2D (*Figure 6—figure supplement 1B*) and rotationally averaged 1D power spectra (*Figure 6—figure supplement 1C*) indicated oscillations beyond 3 Å spatial frequency (*Figure 6—figure supplement 1C*).

All 2D class averages displayed high-resolution detail (*Figure 6A*) and confirmed that FAS was structurally undamaged. This was verified by 3D classification, which showed intact complexes with well-resolved secondary structure. Note that this dataset contained only intact particles (*Figure 6B*), whereas 90% of the particles in the single-particle FAS dataset from unsupported vitrified samples had suffered major damage (*Figure 1B* classes 2, 3), and even the remaining 10% were compromised (*Figure 1B* class one and *Figure 1C*).

After merging the best 3D classes, we obtained a final set of ~28,000 particles. Auto-refinement without symmetry (C1) or with imposed D3 symmetry yielded maps at 4.8 and 4.0 Å resolution, respectively (*Figure 6C,D*). Local resolution estimates indicated better than 3.5 Å resolution for the rigid alpha wheel (*Figure 6—figure supplement 2*). For an unbiased comparison to the 9.5 Å map obtained from FAS in unsupported vitrified buffer (*Figure 6—figure supplement 3A*), we randomly selected 8000 particles from the dataset collected on hydrophilized graphene. The resulting map (*Figure 6—figure supplement 3B*) attained a resolution of 6.4 Å, confirming that hydrophilized graphene works very much better as a substrate for single-particle cryo-EM of FAS than unsupported vitrified buffer. The particles were intact and the map contained all the main features of the best non-symmetrized 4.8 Å map (*Figure 6—figure supplement 3C*). Finally, although not random, particle orientation was much more evenly distributed on hydrophilized graphene, compared to unsupported vitrified buffer (*Figure 6—figure supplement 4*).

## Discussion

An earlier single-particle cryo-EM structure of *S. cerevisiae* FAS in unsupported vitrified buffer (*Gipson et al., 2010*) reported a resolution of 7.2 Å at the 0.5 FSC threshold and

5.9 Å at 0.143 FSC. The 'gold-standard FSC' procedure of estimating map resolution by comparing reconstructions derived from two independent halves of the particle data set (*Chen et al., 2013*; *Scheres and Chen, 2012*) had not been introduced at the time and was not applied. Therefore, the 0.5 FSC resolution estimate of that map was realistic, as confirmed by a comparison of the FAS alpha wheel in the earlier structure to the gold-standard FSC maps in the present study (*Figure 6—figure*

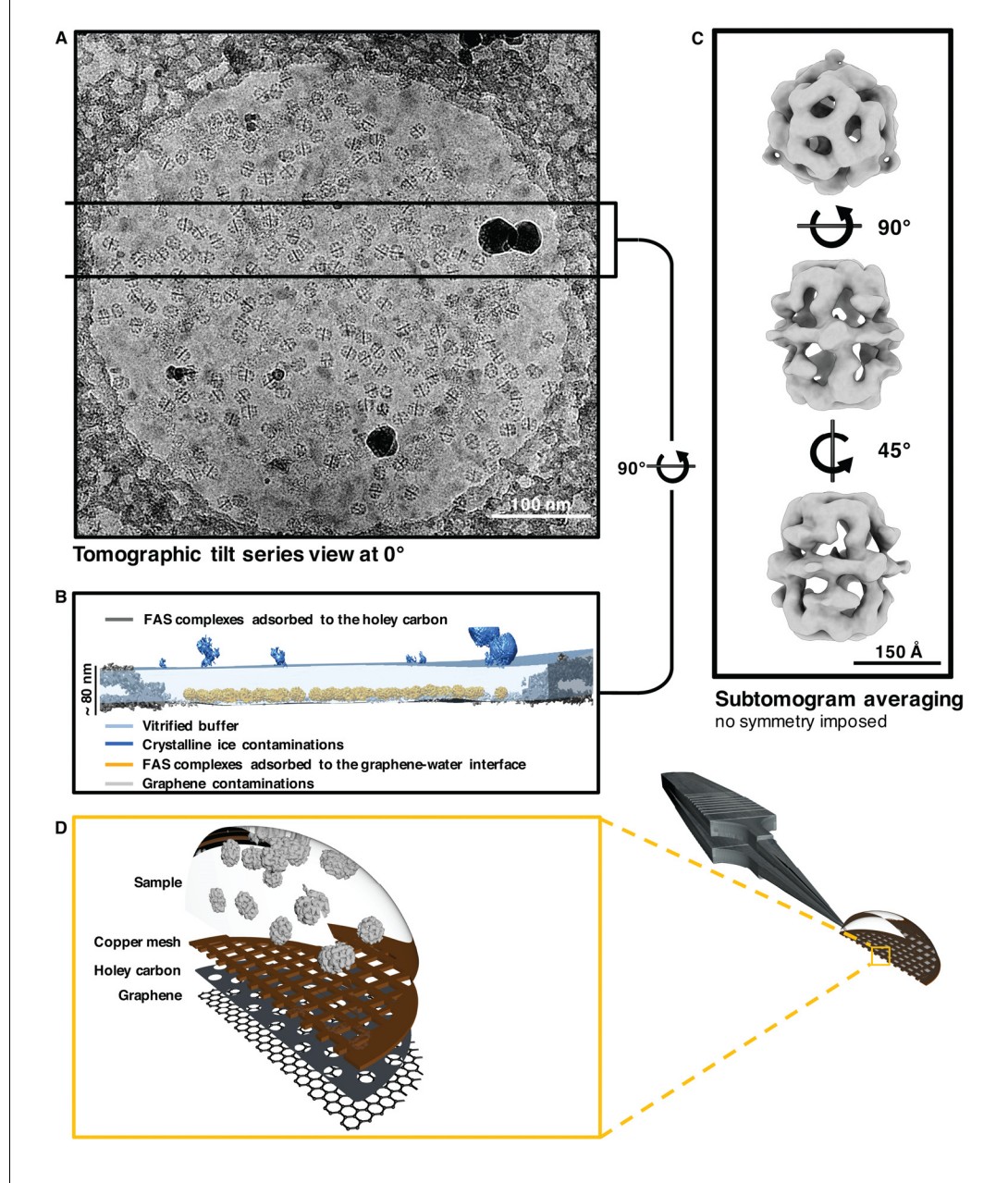

**Figure 5.** Sub-tomogram averaging of FAS vitrified on hydrophilized graphene. (**A**) Zero degree view of tomographic tilt series. (**B**) Slab of vitrified buffer delimited by carbon and ice contaminants, indicating adsorption of FAS complexes to the graphene-water interface. (**C**) Subtomogram averaging confirms the structural integrity of FAS. (**D**) Three-dimensional impression (not drawn to scale) indicating the relative position of Quantifoil carbon film (dark grey) and hydrophilized graphene (mid-grey) on the copper support grid (dark red). The solution containing FAS particles (light grey) was applied from the uncoated side of the grid.

DOI: https://doi.org/10.7554/eLife.42747.011

The following video and figure supplements are available for figure 5:

**Figure supplement 1.** Chemical doping of graphene-coated Quantifoil grids.
DOI: https://doi.org/10.7554/eLife.42747.012

**Figure supplement 2.** Slices through tomographic volume of FAS on hydrophilized graphene.
DOI: https://doi.org/10.7554/eLife.42747.013

**Figure supplement 3.** Angular distribution plot after subtomogram averaging of the particle dataset on hydrophilized graphene support (*Figure 5C*).
DOI: https://doi.org/10.7554/eLife.42747.014

**Figure supplement 4.** Multi-reference alignment of FAS on hydrophilized graphene.

*Figure 5 continued on next page*

*Figure 5 continued*

DOI: https://doi.org/10.7554/eLife.42747.015

**Figure supplement 5.** Resolution estimate of subtomogram averages.

DOI: https://doi.org/10.7554/eLife.42747.016

**Figure 5—video1.** Three-dimensional rendering of FAS on hydrophilized graphene.

DOI: https://doi.org/10.7554/eLife.42747.017

*supplement 5*). The resolution of the earlier map is clearly between that of the 6.4 Å map of FAS on graphene and the 9.5 Å map of FAS in unsupported vitrified buffer, which were obtained with ~8000 particles each. The better quality of the earlier map (*Gipson et al., 2010*) is fully accounted for by the larger number of particles contributing to it, and the application of D3 symmetry. The 4 Å resolution of our present map from 28,000 particles on hydrophilized graphene is most likely limited by the inherent flexibility of the complex. This is implied by the cryo-EM structure of a FAS complex from the thermophilic fungus *Chaetomium thermophilum* (*Kastritis et al., 2017*), which attained a gold standard, FSC 0.143 resolution of 4.7 Å from only ~ 4000 particles. It is well known that protein complexes from thermophilic organisms are more stable than those of mesophilic origin. By comparison, more than 100,000 particles contributed to the 7.5 Å cryo-EM structure of a FAS complex from *Mycobacterium smegmatis* (*Boehringer et al., 2013*), suggesting that the mycobacterial complex is significantly less stable.

In many of the cryo-EM grids examined by electron tomography (*Noble et al., 2018*), the curvatures of the upper and lower meniscus were different. In the case of FAS, the less densely populated upper meniscus was more strongly curved than the lower meniscus (*Figure 2B*, *Figure 2—video 1*). The reason for the difference in curvature is not known and most likely stochastic. The asymmetric distribution of protein on the upper and lower meniscus is surprising, because the grids were blotted symmetrically from both sides. Possibly, the more densely populated lower meniscus remained in contact with air for longer during the blotting process, so that more protein accumulated on it. At this stage, the reason for the asymmetrical particle distribution on the two surfaces is unknown.

Our cryo-EM analysis of FAS, a soluble 2.6 MDa protein complex, revealed that only a minority of the particles in unsupported vitreous films retained intermediate-resolution features, whereas up to 90% were at least partly denatured by the air-water interface. A survey of recently reported high-resolution cryo-EM structures shows that usually only a minor fraction of a large single-particle data set contributes to the final high-resolution map. Percentages of good particles were 19% for the 3.8 Å map of a human synaptic GABAA receptor (*Zhu et al., 2018*); 15% for human P-glycoprotein at 3.4 Å (*Kim and Chen, 2018*); 11.8% for a 4 Å nucleosome map (*Takizawa et al., 2018*); 8.9% for the 3.4 Å structure of human γ-secretase (*Bai et al., 2015b*); and only 5.7% for the 4 Å structure of a sodium channel complex from electric eel (*Yan et al., 2017*). All were prepared in unsupported vitrified buffer. These numbers suggest that up to 94% of the particles may have suffered partial denaturation at the air-water interface. Future studies will show whether this is indeed the case, and whether denaturation can be avoided by using hydrophilized graphene grids, as we have shown for yeast FAS. If hydrophilized or otherwise functionalized graphene grids prove to work as well for other proteins to overcome denaturation at the air-water interface, a much higher proportion of particles would contribute to the final structure. This would result in a large increase in data collection efficiency and significantly better maps. It would be a major boost for cryo-EM.

## Materials and methods

### Strain cultivation and protein purification

Yeast cultures were grown and FAS was purified as previously reported (*Chakravarty et al., 2004*; *Gajewski et al., 2017b*). Haploid FAS-deficient *S. cerevisiae* cells were transfected with plasmids carrying FAS-encoding genes, then grown in YPD medium. After bead disruption and differential centrifugation, the soluble components were purified by strep-Tactin affinity chromatography then by size-exclusion chromatography. The main peak was concentrated to ~4 mg/ml. All purification steps were analyzed by SDS-PAGE.

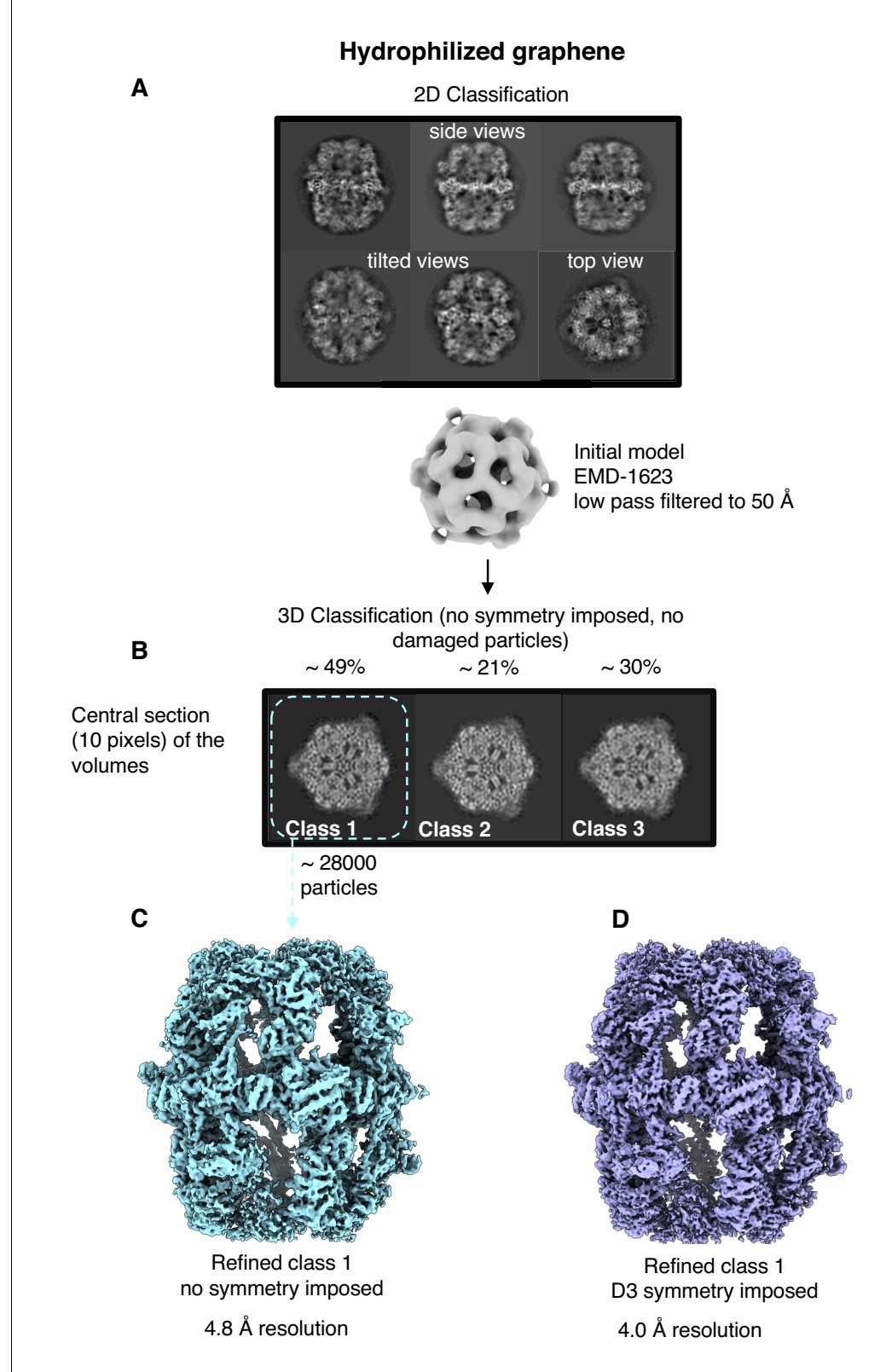

**Figure 6.** Single-particle cryo-EM results on hydrophilized graphene. Two-dimensional (**A**) and three-dimensional (**B**) classification both show intact particles. A final map calculated without (**C**) or with (**D**) imposed D3 symmetry indicated a resolution of 4.8 Å or 4.0 Å.

DOI: https://doi.org/10.7554/eLife.42747.018

*Figure 6 continued on next page*

*Figure 6 continued*

The following figure supplements are available for figure 6:

**Figure supplement 1.** Hydrophilized graphene supported grids.

DOI: https://doi.org/10.7554/eLife.42747.019

**Figure supplement 2.** Local resolution estimates of FAS refined to a global resolution of 4.0 Å (*Figure 6D*) in side view.

DOI: https://doi.org/10.7554/eLife.42747.020

**Figure supplement 3.** Reconstructions from different particle numbers.

DOI: https://doi.org/10.7554/eLife.42747.021

**Figure supplement 4.** Particle orientation distribution.

DOI: https://doi.org/10.7554/eLife.42747.022

**Figure supplement 5.** Comparison of cryo-EM maps obtained in this study to the published cryo-EM map of FAS (*Gipson et al., 2010*).

DOI: https://doi.org/10.7554/eLife.42747.023

## Thermal shift assay (TSA) and activity assay

Thermal shift assays were performed as previously reported. Briefly, 2 µl of protein solution (0.9 mg/ml) were mixed with 21 µl of phosphate buffer (100 mM; pH 6.5) and 2 µl of 62.5 X SYPRO Orange protein gel stain, then fluorescence was measured from 5°C to 95°C with a step of 0.5 °C/min, with excitation wavelength set to 450–490 nm, and emission wavelength to 560–580 nm. FAS activity was determined by tracing NADPH consumption at 334 nm as reported (*Gajewski et al., 2017a*), and adapted for plate reader read-out (120 µl scale containing 200 mM $NaH_2PO_4/Na_2HPO_4$ (pH 7.3), 1.75 mM 1,4-dithiothreitol, 0.03 mg/ml BSA, 0.7 µg FAS, 500 µM malonyl CoA, 417 µM acetyl CoA and 250 µM NADPH).

## Grid preparation

Quantifoil R0.6/1 and R2/2 grids (Quantifoil Micro Tools, Jena, Germany) were used to prepare cryo-specimens with or without graphene support. Grids were washed thoroughly overnight in chloroform. For graphene-coated grids, graphene pads (1 $cm^2$) (Graphenea, Cambridge, MA) were floated onto Quantifoil grids in a water bath. The graphene layer was deposited on the holey carbon side of the grids, whereas the protein sample is later applied on the copper side after mild glow discharge (15 mA for 45 s). Quantifoil R1/2 and R1.2/1.3 grids were tested but found to be less suitable as the smaller hole size yields flatter graphene layers. Grids were dried under nitrogen flow for 30 min and then heated to 150°C for one hour to anneal the graphene layer to the Quantifoil film. Graphene-coated grids were stored under vacuum until use. Graphene-coated grids were washed in pure acetone for one hour to dissolve the protective PMMA layer and then rinsed with isopropanol for another hour, followed by drying under a nitrogen stream. Finally, the grids were dipped into 50 mM 1-pyrenecarboxyilic acid (Sigma Aldrich, Munich, Germany) dissolved in DMSO (Sigma Aldrich, Munich, Germany) for one minute, rinsed in one change of isopropanol and ethanol, and dried under a nitrogen stream. Grid quality was assessed in a FEI Tecnai G2 Spirit BioTwin (FEI Company, Hillsboro, OR) operated at 120 kV, at a nominal magnification of 9300x, yielding a pixel size at the specimen level of 1.19 nm. Electron diffraction patterns were recorded at a nominal camera length of 540 mm, with a 1 s exposure time and 150 µm aperture.

## Negative staining

FAS was diluted in purification buffer (100 mM sodium phosphate, pH 6.5) 0.05 mg/ml and negatively stained with 2% (w/v) sodium silicotungstate (Agar Scientific, Stansted, UK). Specimen preparation was performed as described previously (*Salzer et al., 2016*). Micrographs were recorded in a FEI Tecnai G2 Spirit (FEI Company, Hillsboro, OR) operated at 120 kV, with a pixel size of 2.68 Å.

## Controlled protein denaturation at the air-water interface

Three different experiments of controlled protein denaturation at the air-water interface were carried out in triplicate. Freshly purified FAS solution was diluted to 0.01 mg/ml with purification buffer (100 mM sodium phosphate, pH 6.5). For experiment 1, air was bubbled through 200 µL of protein

solution through a pipette tip for about 10 s, and EM grids were prepared from a 3 µl aliquot. For experiment 2, 200 µl of protein solution were passed over a 5 cm 100 µl intraMARK disposable glass micropipette (Brand, Wertheim, Germany) sealed at both ends, and collected for EM analysis in negative stain. For experiment 3, 20 µl of FAS solution were pipetted onto EM grid coated with amorphous carbon and incubated in air. After 15 s, the drop was touched with a second carbon-coated EM grid and blotted. Both grids were negatively stained as before.

## Single-particle cryo-EM

Three µl of FAS solution (1.0 mg/ml for unsupported grids or 0.2 mg/ml for hydrophilized graphene grids) were applied to freshly glow-discharged Quantifoil R2/2 holey carbon grids (Quantifoil Micro Tools, Jena, Germany) for unsupported samples or Quantifoil R0.6/1 holey carbon grids for hydrophilized graphene. Grids were vitrified in a Vitrobot Mark IV plunge-freezer at 100% humidity and 10°C after blotting for 6–8 s. Cryo-EM images were collected in a Titan Krios (FEI Company, Hillsboro, OR) electron microscope operating at 300 kV. Images were recorded automatically with EPU at a pixel-size of 1.053 Å, on a Falcon III EC direct electron detector (FEI Company, Hillsboro, OR) operating in counting mode. A total of 2648 and 1055 dose-fractionated movies were recorded to a cumulative dose of ~ 32 $e^-/Å^2$ for FAS in unsupported vitrified buffer and on hydrophilized graphene, respectively. Image drift correction was performed using Unblur (*Grant and Grigorieff, 2015*) and MotionCor2 (*Zheng et al., 2017*). CTF determination with CTFFIND 4.1.10 (*Rohou and Grigorieff, 2015*). All subsequent image processing steps were performed within Relion 2.1 (*Kimanius et al., 2016*). An initial dataset of 81,163 unsupported particles was automatically picked, 2D classified, and used for the first consensus refinement with a 50 Å low pass filtered FAS EM map (EMD-1623 (*Gipson et al., 2010*)) as initial reference. No symmetry was imposed. The same procedure was applied to the data collected on hydrophilized graphene grids, starting from a dataset of 57,021 particles. The best particles, sorted by 3D classification, were combined to perform a reconstruction imposing C1 (no symmetry) or D3 symmetry. This yielded maps at 9.5 Å resolution (unsupported samples, C1 symmetry), 4.8 and 4.0 Å resolution (hydrophilized graphene, C1 and D3 symmetry, respectively). All maps were corrected for the modulation transfer function (MTF) of the detector and sharpened with a B-factor of −130 $Å^2$. Particle back-tracking was performed with the script *star2jpg* (https://github.com/olibclarke/EM-scripts/blob/master/star2jpg.bash).

## Electron cryo-tomography

Vitrified specimens were imaged in a Titan Krios (FEI Company, Hillsboro, OR) electron microscope operating at 300 kV, equipped with a K2 summit direct electron detector and a Quantum energy filter (Gatan, Inc., Pleasanton, CA). The magnification was set to a pixel size of 3.39 Å or 2.20 Å for the dataset in unsupported aqueous films and hydrophilized graphene, respectively. Dose-fractionated tomographic images were automatically recorded in counting mode from −60° to + 60° using a dose-symmetric acquisition scheme (*Hagen et al., 2017*) implemented in SerialEM (*Mastronarde, 2005*) to a cumulative dose of 90 $e^-/Å^2$ per tilt series. After movie frame alignment with MotionCor2 (*Zheng et al., 2017*) and CTF correction with Gctf (*Zhang, 2016*), the image stack files were processed with IMOD (*Kremer et al., 1996*). Weighted back projection was used to reconstruct the 3D volumes after patch track alignment. If necessary, the tomograms were processed with a Nonlinear Anisotropic Diffusion (NAD) filter (*Frangakis and Hegerl, 2001*) for visualization.

## Subtomogram averaging, volume segmentation and rendering

All processing steps were performed with Dynamo (*Castaño-Díez, 2017*). A set of 1724 and 2090 particles was manually picked from unsupported and hydrophilized graphene-coated grids, respectively. In both cases a subset of 20 random particles was used to generate an initial reference, and subtomogram averaging was performed according to the 'gold standard' procedure (*Scheres and Chen, 2012*). The final map was band-pass filtered to 308 and 12 Å. To assess mask bias, FSC was also performed on the masked half-maps with phases randomized beyond 60 Å. The correlation dropped at the resolution above which the phases were randomized, indicating that the mask did not affect the resolution estimate. Finally, particle heterogeneity was explored by multi-reference alignment. To exclude reference bias, average volumes of damaged and intact FAS complexes were

used as initial models for the graphene and the dataset unsupported dataset, respectively. Both references were low-pass filtered to 50 Å. For illustrative purposes, final maps were Gaussian-filtered (standard deviation of two physical pixels) within UCSF Chimera (*Pettersen et al., 2004*) and tomographic volumes segmented with the convolutional neural network method implemented in EMAN2.2 (*Chen et al., 2017*; *Tang et al., 2007*).

### Estimation of particle-to-interface orientation

To determine the orientation of partly denatured FAS with respect to the air-water interface, MATLAB was used to correlate the tomographic reconstruction with a geometrical model assuming that, upon adsorption, the plane describing the denatured side of the FAS particles would be parallel to the air-water interface. The missing density due to denaturation ($R_{missing}$) was treated as the difference between the reconstruction of intact FAS ($R_{intact}$) and the map obtained from sub-tomogram averaging of denatured particles ($R_{denat}$). The analysis consisted of five sequential steps: (i) coordinates of the center of FAS complexes previously determined by sub-tomogram averaging were used to model the air-water interface ($S_{estimate}$) by the *Thin-plate interpolator* option of the *Curve Fitting* Toolbox in MATLAB; (ii) for each particle location the vector $P_{normal}$ was calculated, which represents the normal of $S_{estimate}$ at that position; (iii) a vector $P_{denat}$ was computed that describes the relative orientation of the denatured side as the vector pointing from the center of $R_{intact}$ to the center of mass of $R_{missing}$; (iv) the $P_{denat}$ vector was calculated for every particle detected in all the tomographic volumes; (v) finally, the displacement angle δ between the vectors $P_{denat}$ and $P_{normal}$ was calculated with a 7.5° sampling step. The distribution of δ was plotted as a bild file. All the 3D rendering and movie editing were performed with Blender, Chimera (*Pettersen et al., 2004*) and ChimeraX (*Goddard et al., 2018*).

### Data and materials availability

The EM maps have been deposited in the EMDB with accession codes EMD-0178 (single particle cryo-EM FAS map on graphene support) and EMD-0179 (subtomogram averaging FAS map on graphene support).

## Acknowledgements

We thank Deryck J. Mills, Simone Prinz and Mark Linder for EM support. We are grateful to Martin Centola, Niklas Klusch and Dr. David Wöhlert for discussions. We thank Dr. Janet Vonck, Dr. Roberto Covino and Dr. Mikhail Kudryashev for critically reading the manuscript.

## Additional information

#### Competing interests

Werner Kühlbrandt: Reviewing editor, *eLife*. The other authors declare that no competing interests exist.

#### Funding

| Funder | Grant reference number | Author |
|---|---|---|
| Max-Planck-Gesellschaft | | Edoardo D'Imprima |
| Max-Planck-Gesellschaft | | Davide Floris<br>Ricardo Sánchez |
| Volkswagen Foundation | | Mirko Joppe |
| Alexander von Humboldt Foundation | Sofja Kovalevskaja Award to Mikhail Kudryashev | Ricardo Sánchez |
| Deutsche Forschungsgemeinschaft | SFB807 | Ricardo Sánchez |
| Volkswagen Foundation | 85701 | Martin Grininger |

The funders had no role in study design, data collection and interpretation, or the decision to submit the work for publication.

## Author contributions

Edoardo D'Imprima, Conceptualization, Data curation, Formal analysis, Supervision, Validation, Investigation, Visualization, Methodology, Writing—original draft, Project administration, Writing—review and editing; Davide Floris, Data curation, Formal analysis, Validation, Investigation, Visualization, Methodology, Writing—original draft, Writing—review and editing; Mirko Joppe, Formal analysis, Methodology, Writing—review and editing; Ricardo Sánchez, Software, Formal analysis, Validation, Methodology, Writing—review and editing; Martin Grininger, Werner Kühlbrandt, Resources, Supervision, Funding acquisition, Writing—review and editing

## Author ORCIDs

Edoardo D'Imprima (iD) http://orcid.org/0000-0002-9830-7929
Davide Floris (iD) http://orcid.org/0000-0001-9144-5459
Mirko Joppe (iD) http://orcid.org/0000-0001-6463-0253
Werner Kühlbrandt (iD) https://orcid.org/0000-0002-2013-4810

## Decision letter and Author response

Decision letter https://doi.org/10.7554/eLife.42747.030
Author response https://doi.org/10.7554/eLife.42747.031

# Additional files

## Supplementary files

• Transparent reporting form
DOI: https://doi.org/10.7554/eLife.42747.024

## Data availability

The EM maps have been deposited in the EMDB with accession codes EMD-0178 (single particle cryo-EM FAS map on graphene support) and EMD-0179 (subtomogram averaging FAS map on graphene support).

The following datasets were generated:

| Author(s) | Year | Dataset title | Dataset URL | Database and Identifier |
|---|---|---|---|---|
| D'Imprima E, Davide Floris | 2018 | Cryo-EM map of the Fatty Acid Synthase from S. cerevisiae | http://www.ebi.ac.uk/pdbe/entry/emdb/EMD-0178 | Electron Microscopy Data Bank, EMD-0178 |
| D'Imprima E, Davide Floris | 2018 | Cryo-EM map of the Fatty Acid Synthase from S. cerevisiae | http://www.ebi.ac.uk/pdbe/entry/emdb/EMD-0179 | Electron Microscopy Data Bank, EMD-0179 |

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
