## [Decision Letter]

Thank you for submitting your article "Protein denaturation at the air-water interface and how to prevent it" for consideration by *eLife*. Your article has been reviewed by three peer reviewers, including Axel Brunger as Reviewing Editor, and the evaluation has been overseen by John Kuriyan as the Senior Editor. The following individuals involved in review of your submission have agreed to reveal their identity: Robert M Glaeser (Reviewer #2); Georgios Skiniotis (Reviewer #3).

The reviewers have discussed the reviews with one another and the Reviewing Editor has drafted this decision to help you prepare a revised submission.

Summary:

This study explores the effects of adsorption to the air-water interface on protein particle structure and orientation. Using both single-particle cryo-EM and cryo-ET coupled to sub-tomogram averaging, the authors provide a thorough topological and structural analysis of vitrified protein particles using the well characterized yeast fatty acid synthase (FAS) as a model system. This work follows the recent cryo-ET study by Carragher and colleagues showing that most particles prepared with the typical vitrification process preferentially adsorb to the air-water interface with potentially detrimental effects to their structure. Here, D'Imprima and colleagues build on those findings and demonstrate convincingly that up to ~90% of particles residing at the interface have suffered damage through denaturation, with the denatured region facing the interface. The cryo-EM analysis, coupled to simple but very elegant negative stain imaging of particle integrity under different conditions, clearly reveals that particles have been damaged during vitrification and that this is associated with exposure to air; an event that accounts for a large fraction of discarded particles with important implications for data collection and structural analysis. The authors go further to demonstrate that the use of hydrophilized (through non-covalent doping) graphene prevents FAS denaturation and allows the calculation of highly improved 3D reconstructions.

Essential revisions:

The authors propose the use of hydrophilized graphene oxide as a general solution to avoid protein denaturation during vitrification. To make such a strong impact however, this work should include a demonstration of the general use of this support with additional and different types of samples. In the absence of such additional experiments, the paper may be acceptable only if it focuses its message on yeast FAS. Statements in the title, Abstract and throughout the paper about the general application of this method should be removed.

The treatment by 1-pyrCA is interesting. In previous publications (e.g., Russo and Passmore, 2014), hydrogen-plasma treatment was used. Did the authors try this method? If so, how does it compare with the 1-pyrCA method?

It is common that papers about using graphene or graphene oxide as support films claim that such specimen-support films give much less background noise in the images than is the case for evaporated carbon support films. While that is clearly expected to be true, in this case the data seem to show that the background noise is much higher after the functionalization steps than it was before, and the authors properly point this out. The authors should either not make such a big point at the beginning about there being a low-noise background (i.e. do not raise false expectations), or point out, whenever making such statements, that the FGM reported here do not fully deliver the desired low-noise support film.

• It is not enough to suggest, as the authors do, that the increased RMS background is similar in height to that of the functionalized chemical groups that were added.

• Nor is it enough to point out, as the authors might do, that the noise did not interfere with getting a high-resolution map of FAS, because this quite a large particle.

• The community will still be skeptical that the same type of grids will be useful for particles in the size range of 200 kDa or smaller.

The manuscript does not fully represent the extent to which the cryo-EM community now appreciates that the air-water interface can do bad things. The unintended consequence is that the manuscript appears to claim more credit for originality than is perhaps due.

• A simple change might be to remove the statement, at the end of the first paragraph of the Introduction, which says "Effects of adsorption.… are not widely appreciated."

• A more extensive change would be to add a more comprehensive review of recent literature.

Please clarify which surface it is, in Figure 2B, to which the sample was applied. When sample is applied to a holey carbon grid, there are small disks of air-water interface within each hole and a large, continuous air-water interface on the top side of the sitting drop of sample. Of these two air-water interfaces, which is the one that has so many particles adsorbed to it?

It would be informative if the authors would describe whether bubbling air through their sample, in the way that they described, resulted in foam, and if so, how long it persisted. Indeed, a photo of the foam (if there was any) would be a good addition to the supplementary material.

The discussion on the lower and upper meniscus of the vitrified layer requires further explanation and clarification. What is their relation to the direction the sample is applied?

In reference to the 2010 cryo-EM study, the authors state that "The fact that the resolution was limited to 7.2 Å even with more than twice the number of particle images suggests that the complex was equally affected by denaturation at the air-water interface". This statement needs correction. The resolution was primarily limited by recording on film without the ability to correct for specimen motion.

There are particles in Figure 4D that do appear half, not unlike damaged particles at the interface, although they have adsorbed to the carbon. How do the authors explain these types of particles, also compared to the ones in the typical experiment of 4A?

---

## [Author Response]

Essential revisions:The authors propose the use of hydrophilized graphene oxide as a general solution to avoid protein denaturation during vitrification. To make such a strong impact however, this work should include a demonstration of the general use of this support with additional and different types of samples. In the absence of such additional experiments, the paper may be acceptable only if it focuses its message on yeast FAS. Statements in the title, Abstract and throughout the paper about the general application of this method should be removed.

First, the reviewers seem to be under the impression that we used graphene oxide to coat the grids (see first line of the reviewer comments), we would like to clarify that we used only pristine graphene.

Second, we did not propose anywhere in the manuscript that the use of hydrophilized graphene is a general solution to the problem of protein denaturation. On the contrary, we were careful not to make such a claim. In this respect, we explicitly say that future studies will have to show that our method works as well for other proteins, and if it does, it will be a major boost. The final paragraph of the Discussion reads:

“These numbers suggest that up to 94% of the particles may have suffered partial denaturation at the air-water interface. […] This would result in a large increase in data collection efficiency and significantly better maps. It would be a major boost for cryo-EM.”

Nevertheless, to avoid any appearance of making such a claim, we added “potentially” to the sentence “As a simpler and potentially more general solution, we propose to use a physical support that largely prevents protein contact with, and consequently denaturation at, the air-water interface.”

We modified the sentence “We used fatty acid synthase (FAS) from *Saccharomyces cerevisiae* to explore the denaturing effect of the air-water interface and how to avoid it.” with “We explored the denaturing effect of the air-water interface on fatty acid synthase (FAS) from *Saccharomyces cerevisiae* as an example of a large protein complex and devised a way to avoid it.”

We modified the sentence “Finally, we demonstrate by high-resolution single-particle cryo-EM that a stable substrate of hydrophilized graphene avoids the denaturation during cryo-EM specimen preparation completely.” with “Finally, we demonstrate by high-resolution single-particle cryo-EM that a stable substrate of hydrophilized graphene avoids denaturation of FAS complex during cryo-EM specimen preparation completely.”

The treatment by 1-pyrCA is interesting. In previous publications (e.g., Russo and Passmore, 2014), hydrogen-plasma treatment was used. Did the authors try this method? If so, how does it compare with the 1-pyrCA method?

We did not have a hydrogen plasma cleaner at the time when this work was conducted. One important advantage of our method is that it does not depend on such a device, which is expensive and not available in many cryoEM laboratories.

It is common that papers about using graphene or graphene oxide as support films claim that such specimen-support films give much less background noise in the images than is the case for evaporated carbon support films. While that is clearly expected to be true, in this case the data seem to show that the background noise is much higher after the functionalization steps than it was before, and the authors properly point this out. The authors should either not make such a big point at the beginning about there being a low-noise background (i.e. do not raise false expectations), or point out, whenever making such statements, that the FGM reported here do not fully deliver the desired low-noise support film.• It is not enough to suggest, as the authors do, that the increased RMS background is similar in height to that of the functionalized chemical groups that were added.• Nor is it enough to point out, as the authors might do, that the noise did not interfere with getting a high-resolution map of FAS, because this quite a large particle.• The community will still be skeptical that the same type of grids will be useful for particles in the size range of 200 kDa or smaller.

We agree that our images are noisier than one would expect for pristine graphene (not graphene oxide). While the contrast we obtained is significantly better than for standard evaporated carbon support films, we cannot exclude some extent of hydrocarbon contamination before or after doping. After submitting our manuscript, we found that 1-pyrCA renders graphene hydrophilic at a very much lower concentration, i.e. in the nanomolar range. The millimolar concentration of 1-pyrCA reported in the manuscript may have contributed to background noise in the images.

We modified statements in the text which may have raised unrealistic expectations regarding our chemical doping method. Specifically, we modified “The advantage of non-covalent doping is that the pristine graphene surface is preserved, and that particle adsorption can be tuned by adjusting the concentration of the doping chemical.” to “The advantage of non-covalent doping is that the mechanical and chemical properties of graphene surface are preserved, and that particle adsorption can be tuned by adjusting the concentration of the doping chemical.”. We exchanged “Typical micrographs recorded at 0.9 μm defocus showed good contrast (Figure 6—figure supplement 1A).” with “Typical micrographs recorded at 0.9 μm defocus showed well-preserved particles (Figure 6—figure supplement 1A), although image contrast was not as good as expected. Presumably the pristine graphene surface became contaminated to some extent with atmospheric hydrocarbons during specimen preparation, and the high dopant concentration may contribute to some loss of contrast. These factors may compromise the detection and alignment of particles that are significantly smaller than yeast FAS.”

The manuscript does not fully represent the extent to which the cryo-EM community now appreciates that the air-water interface can do bad things. The unintended consequence is that the manuscript appears to claim more credit for originality than is perhaps due.• A simple change might be to remove the statement, at the end of the first paragraph of the Introduction, which says "Effects of adsorption.… are not widely appreciated."• A more extensive change would be to add a more comprehensive review of recent literature.

We rephrased the statement “Effects of adsorption to the air-water interface on protein integrity, orientation, and structure have not been investigated in detail and are not widely appreciated.” to “Recent studies (Glaeser, 2018; Glaeser and Han, 2017; Han, Watson, Cate, and Glaeser, 2017) have drawn attention to the effects of the air water interface on proteins in solution, in particular on their integrity and orientation cryoEM grids.”

Please clarify which surface it is, in Figure 2B, to which the sample was applied. When sample is applied to a holey carbon grid, there are small disks of air-water interface within each hole and a large, continuous air-water interface on the top side of the sitting drop of sample. Of these two air-water interfaces, which is the one that has so many particles adsorbed to it?

We have now tracked the grid orientation from sample application to subtomogram averaging carefully and can confirm that the densely populated meniscus is on the lower side of the drop, i.e. on the side opposite from where the sample was applied. This agrees with our thoughts that the more populated meniscus had been exposed to air for longer than the fresh meniscus that forms upon blotting. It also explains why we did not detect damaged particles in graphene supported samples, since the “lower meniscus” (as in Figure 2B) is protected by a graphene layer (as shown in Figure 5D).

It would be informative if the authors would describe whether bubbling air through their sample, in the way that they described, resulted in foam, and if so, how long it persisted. Indeed, a photo of the foam (if there was any) would be a good addition to the supplementary material.

In our experiments, air bubbling did not produce foam.

The discussion on the lower and upper meniscus of the vitrified layer requires further explanation and clarification. What is their relation to the direction the sample is applied?

The upper meniscus, mentioned in Figure 2B, represents the side from which the sample was applied. Please see the comment above.

In reference to the 2010 cryo-EM study, the authors state that "The fact that the resolution was limited to 7.2 Å even with more than twice the number of particle images suggests that the complex was equally affected by denaturation at the air-water interface". This statement needs correction. The resolution was primarily limited by recording on film without the ability to correct for specimen motion.

Sentence removed.

There are particles in Figure 4D that do appear half, not unlike damaged particles at the interface, although they have adsorbed to the carbon. How do the authors explain these types of particles, also compared to the ones in the typical experiment of 4A?

When the droplet surface is touched with a second grid (upper tweezers in Figure 4C), it is difficult to control the exact amount of liquid remaining on the first grid (lower tweezers). When the film becomes too thin, all FAS particles in the small remaining volume are at risk from damage at the air-water interface. Most likely the few damaged particles in Figure 4D became denatured in the thin film of solution before stain was applied.